# Fundamental Technologies and Recent Advances of Cell-Sheet-Based Tissue Engineering

**DOI:** 10.3390/ijms22010425

**Published:** 2021-01-03

**Authors:** Chikahiro Imashiro, Tatsuya Shimizu

**Affiliations:** Institute of Advanced Biomedical Engineering and Science, Tokyo Women’s Medical University, Tokyo 162-8666, Japan; imashiro.chikahiro@twmu.ac.jp

**Keywords:** cell sheet technology, tissue engineering, regenerative therapy, tissue maturation, vasculogenesis

## Abstract

Tissue engineering has attracted significant attention since the 1980s, and the applications of tissue engineering have been expanding. To produce a cell-dense tissue, cell sheet technology has been studied as a promising strategy. Fundamental techniques involving tissue engineering are mainly introduced in this review. First, the technologies to fabricate a cell sheet were reviewed. Although temperature-responsive polymer-based technique was a trigger to establish and spread cell sheet technology, other methodologies for cell sheet fabrication have also been reported. Second, the methods to improve the function of the cell sheet were investigated. Adding electrical and mechanical stimulation on muscle-type cells, building 3D structures, and co-culturing with other cell species can be possible strategies for imitating the physiological situation under in vitro conditions, resulting in improved functions. Finally, culture methods to promote vasculogenesis in the layered cell sheets were introduced with in vivo, ex vivo, and in vitro bioreactors. We believe the present review that shows and compares the fundamental technologies and recent advances for cell-sheet-based tissue engineering should promote further development of tissue engineering. The development of cell sheet technology should promote many bioengineering applications.

## 1. Introduction

Tissue engineering has attracted significant attention since the 1980s. Originally, tissue engineering represented the field of fabricating implantable tissue, using cells, for clinical therapy. Thus, the expected application of tissue engineering was regenerative medicine when this concept was developed. However, the applications of tissue engineering have expanded to include fundamental research in biology, drug development, and currently even cultured meat [1,2], thereby implying an increasing value of tissue engineering in this frequently changing world.

To develop fundamental technologies for tissue engineering, many researchers have performed trials to construct tissues from cells. In most of the trials, cells were seeded into a three-dimensional (3D) scaffold for the fabrication of 3D tissue, resulting in cell-sparse tissue fabrication [3]. Hence, to fabricate a cell-dense tissue, an alternative method was required. Cell sheet technology, thus, has been studied as a promising strategy to produce a cell-dense tissue [4]. Cell sheet technology is one of the tissue engineering methodologies previously established by our research institute. This technology is based on a “cell sheet”, which is a sheet-like aggregation of cells having an extracellular matrix (ECM). Owing to these features of having the shape of a sheet and ECM, cell sheets are easy to handle and have strong adhesion properties. Thus, cell sheets are easy to transplant, and 3D cell-dense tissues can be fabricated by layering multiple cell sheets.

Although the usability of cell sheets resulted in the development of cell sheet technology for several applications, including the generation of organ models and the fabrication of bio-actuators, fundamental research involving tissue engineering is mainly introduced in this review [5,6,7,8]. While clinical research is being conducted, fundamental studies on cell sheet technology have been mainly conducted and described for the further development of tissue engineering. In this review, the studies are classified into those describing the cell sheet fabrication methods and methods for improving the function of cell sheets. Note that the papers using temperature-responsive culture surfaces were mainly reviewed for the experiments conducted to improve the function of cell sheets. Trials performed to achieve perfusable blood vessel structure technology are individually reviewed, since it might be a critical breakthrough technology in cell sheet technology. Figure 1 describes each chapter, reviewing the overall cell sheet technologies.

## 2. Cell Sheet Fabrication Technologies

When cells reach confluence, they adhere to each other and develop a confluent monolayer on a culture surface, with tight adhesions between them. Generally, an enzyme such as trypsin is employed to harvest cells, which results in dispersed single cells, since adhesion proteins are non-specifically destroyed. At this stage, cells detach in a state of cell sheet if adhesions between the cells and culture surface are selectively lost (Figure 2). Although the temperature-responsive polymer-based technique triggered the establishment and spread of cell sheet technology, other technologies for cell sheet fabrication have also been reported. In this review, cell sheet fabrication technologies are classified into the following three categories: temperature-responsive polymer-based methods, other surface-modification-based methods, and methods that do not use surface modifications. Each has its own benefits and may have appropriate applications as well. Figure 3 shows a summary of each method, and Table 1 shows the advantages and limitations of each method.

### 2.1. Temperature-Responsive Culture Surface

Temperature-responsive culture surfaces were developed for cell detachment, which triggered the spread of cell sheet technology [9]. At a lower critical solution temperature of 32 °C, the temperature-responsive polymer, poly (*N*-isopropylacrylamide) (PIPAAm), can undergo a distinct transition from hydrophobic to hydrophilic. Thus, PIPAAm was immobilized covalently on the surface of ubiquitous culture vessels, which enabled the control of cell adhesion and detachment with changes in temperature. While cell–cell adhesions remain unaffected, cell–surface adhesions are lost. Thus, cultured cells can be detached as a cell sheet by exposing them to a reduced incubation temperature. Cell sheets detached from temperature-responsive culture surfaces have been used in regenerative therapy studies. Myoblast cell sheets were used as the first trial in medical practice that was based on cell-sheet-based regenerative therapy [10]. Further, many clinical and in vivo studies have been reported to date [11,12,13].

Although culture vessels with temperature-responsive culture surfaces became commercially available, methods to further modify the surfaces have been studied. Studies on generating more suitable temperature-responsive culture surfaces or immobilizing temperature-responsive polymers on several surfaces have been performed [14,15,16]. Optimization of temperature-responsive culture surfaces has been reported to improve the efficiency of cell-culture and cell-sheet detachment. Moreover, the thickness of the temperature-responsive polymer on culture surfaces affects cell adhesion; while thicker layers never show cell adhesion, thinner ones fail to detach the adhered cells [17]. Thus, the temperature-responsive polymer thickness has been optimized to achieve efficient cell culture and cell sheet detachment. Mechanically stretchable temperature-responsive culture surfaces were fabricated, and it became possible to vary the thickness of the temperature-responsive polymer by mechanical stretching. In short, cell adhesion strength can be regulated by stretching. The key result of this study is the development of a novel technique to immobilize a temperature-responsive polymer on a polydimethylsiloxane (PDMS) substrate [18]. On the other hand, a stable method for immobilizing the temperature-responsive polymer on a culture insert by spin coating was reported [16]. Cell sheet detachment from culture inserts contributes to the development of tissue engineering, since cells cultured on the inserts were provided with more culture medium, which enhanced cell viability [19]. Owing to these technologies, the applicability of cell species that can be used for cell sheet technology has expanded. Further, not just PIPAAm, but several others polymers or combinations such as poly(N-isopropylacrylamide-co-2-carboxyisopropylacrylamide), etc., have also been developed, which has also contributed to cell sheet technology, as well [20,21,22].

### 2.2. Other Surface-Modification Technologies

Although temperature-responsive culture surfaces have been proven to be widely effective, other surface-modification technologies have also been studied.

#### 2.2.1. Gold-Coated Surface

The formation of gold–thiolate bonds on gold-coated surfaces was reported as an immediate detaching method [23,24]. An electro-responsive cell-sheet detaching system was developed by covalent bonding of Arg–Gly–Asp peptides to alkanethiol molecules. Cell sheet detachment was caused by the release of peptides from the gold surface, using electrical stimulation. With fibroblast cell sheets, it was shown that 10 min of applying −1.0 V electrical potential to the surface allowed cell sheets to detach quickly. Note that single-cell detachment was attained within 2 min. To adopt this technology for tissue engineering and regenerative medicine, further studies have been performed. By applying this method to the surface of cell culture inserts, thicker fibroblast cell sheets can be detached than those cultured on culture dishes where PIPAAm was immobilized [25]. This is because the use of cell culture inserts increased the diffusion coefficient of oxygen and nutrition in the media, thereby promoting cell growth.

Further, a technique for fabricating cell sheets with tailor-made 3D shapes has been exploited to employ cell sheets detached by using an electro-responsive cell-sheet detaching system for direct transplantation [26]. To achieve tailor-made 3D shapes, an electro-responsive cell-sheet detaching surface was developed on the surface of 3D-printed objects, thereby demonstrating the fabrication of a cell sheet with a 3D shape. This type of cell sheet may be useful for regenerating intestines or other organs with complex structures. Furthermore, human neonatal skin fibroblast sheets were transplanted into nude mice, to confirm the biocompatibility of the cell sheet detached by using this method. The advantage of this method is the implementation of 3D-shaped cell sheets.

#### 2.2.2. Photo-Responsive Material

Light illumination has been reported as an option to control cell adhesion because of its good temporal and spatial controllability [27]. Many cell-sheet detachment methods utilizing light-responsive systems have been reported. While some of them directly regulate cell adhesion, there are methods that employ the photothermal effect to dissociate a layer between adhesion protein and a culture surface.

In direct methods, light-responsive materials whose wettability or electrical charge can be controlled by light stimulus are employed for harvesting cell sheets that cover culture surfaces. Metal oxides, especially zinc oxide and titanium dioxide, have been studied for the application of changes in wettability that occur with exposure to ultraviolet (UV) [28]. Exposure to UV might affect cell functions, which is a potential concern. On the other hand, the electrical charge of culture surfaces can be controlled by UV or even via visible light exposure, and cell sheets can be harvested because of the changed electrical charge [29]. This is because, when the electrical charge changes from negative to positive, electrostatic repulsion occurs between the cell membranes and culture surfaces, both with a positive charge.

On the other hand, photo-responsive methods using a photothermal system have been reported. These methods utilize an increase in temperature due to the photo-absorbance property of poly (3,4-ethylenedioxythiophene) (PEDOT) materials [27,30,31]. Large-area living cell sheets were obtained with a detached area larger than 19 cm^2^. In this report, the PEDOT surface was coated with collagen, which is disassembled by photothermal heat. Thus, photo exposure disassembles collagen through its photothermal effect, thereby resulting in cell-sheet detachment. One benefit of photo-responsive methods using PEDOT is the reusability of the PEDOT surface [32].

#### 2.2.3. Materials Inducing Reactive Oxygen Species (ROS) Production

Originally, materials inducing the ROS response were used for single-cell detachment [33]. It is known that cell adhesion is inhibited by an increase in intracellular ROS levels, thereby resulting in cell detachment. Furthermore, it was revealed that extracellular ROS levels also cause cell detachment. Additionally, messenger RNA (mRNA) quantification results revealed that the expression of adhesion-related integrin was reduced due to an increase in ROS levels [34]. However, cell-sheet detachment did not occur with this alteration. Therefore, as a new strategy to detach cell sheets, ROS-responsive methods based on hematoporphyrin-incorporated polyketone films (Hp-PK films), which are fabricated by spin coating, have been reported [35]. The viability of cells cultured on Hp-PK film was confirmed. In this method, after irradiating Hp-PK film with green light-emitting diode (LED) (510 nm), exogenous ROS were produced that induced cell detachment. LED irradiation did not change the wettability of the Hp-PK film but changed the protein structure, which may be one of the triggers of cell detachment. Cells cultured on Hp-PK film are easy to handle because the film has good mechanical properties. Moreover, to transfer cell sheets from Hp-PK films to target sites, a cell monolayer cultured on Hp-PK film was placed on the target sites and then exposed to LED light. In the final step of the one-step cell sheet transfer method, the Hp-PK film was removed gently after cell sheet detachment. Unlike other methods, when the cell sheet is attached to the target, the adhesive surface is limited to the top side, so that the polarity of the cell sheet is turned upside down. In vivo experiments using nude mice were also performed, to demonstrate the potential of this method in the field of regenerative medicine. The advantage of this method is the ease of transplantation.

### 2.3. Technologies other Than Surface Modification

While several surface modification technologies for cell sheet fabrication have been reported, technologies that do not require any surface modification have also been developed.

#### 2.3.1. Dispase Treatment

Dispase treatment was the first method used to fabricate cell sheets, which was also the first report of regenerative medicine employed in humans. Autologous human epidermal cells were cultured to form epithelial skin grafts in vitro and could be used to repair defects in the epidermis in cases such as burns [36]. After cells reached confluence on the culture surface, a cell sheet was detached from the culture surface by treating the cells with 1% dispase for 15 min at 37 °C and using a cell scraper. Although the survival rate of dispase-treated cell sheets after transplant was lower than that of the cell sheets detached from temperature-responsive polymers due to the digested ECM, a dispase-based method is still used to fabricate cell sheets, especially for fabricating epithelial cell sheets [37]. Further, fundamental studies on dispase-treated cell sheets are being performed [38]. This method allows the cell sheet to be detached without involving any non-physiological stimulation process. Further, as another method using enzyme, it has been reported that collagenase-treated oral mucosal epithelial cell sheets, which have already been predicted to be effective in vivo can be harvested [39,40].

#### 2.3.2. Fibrin Matrix Coating

As a simple method of fabricating cell sheets, a fibrin-based method was reported immediately after the development of the dispase-based method. A homogenous fibrin matrix is formed on the surface of a Petri dish by spreading a mixture of fibrin glue [41]. Subsequently, the cells were seeded on the matrix. After culturing the cells, the cell monolayer was gently detached from the Petri dish, using two forceps, and the cell sheet was then fabricated using fibrin glue. Because of fibrin glue, the cell sheet detached by using this method provides long-term durability for handling and good adhesion properties, which are the advantages of using this method.

Furthermore, a similar method using fibrin glue was reported. The study demonstrated that, although a homogenous fibrin matrix is formed, fibrin is digested by cells themselves [42]. Some cells such as myocardial cells secrete proteases that digest fibrin; therefore, during cell culture, the fibrin matrix supporting cell adhesion to the culture surface is digested. Therefore, because of the digested fibrin layer, the cell sheet becomes available for electrical connection to other cell sheets. The action potential of the myocardial cell sheet detached by using this method propagated to another cell sheet on the cell sheet without any delay, which was similar to that of the cell sheet detached from a temperature-responsive culture surface. This electrical connection is an advantage identified in this study, because it may reduce the risk of arrhythmia caused by unsynchronized spontaneous beating after the transplantation of myocardial cell sheets.

#### 2.3.3. Magnetic Nanoparticles

Magnetic nanoparticles, which are utilized for treating cells, have been used for cell sheet detachment [43]. It is known that magnetically labeled cells can be harvested with magnets and have demonstrated biocompatibility. In cell sheet fabrication, the magnetically labeled cells are seeded on ultralow-attachment surfaces. Subsequently, to fabricate a cell sheet-like shape, a magnet is placed under the surface. After culturing the cells for a certain duration, the magnet was removed, and the cell sheet was detached from the ultralow-attachment surface. Owing to this external force, the thickness of the cell sheet formed on the culture surface can be easily regulated by controlling the number of seeded cells. In other methods, cell sheets are stacked to fabricate 3D cell tissues; therefore, the ability to control the thickness of the formed cell sheets should be an advantage of this method.

Although magnetic nanoparticles were added to the culture environment, the particles demonstrated good biocompatibility from the viewpoints of cell viability and cell adhesion. Further, the potential of this method in regenerative therapy was confirmed by in vivo experiments conducted in several cell species [44].

#### 2.3.4. Ultrasound Irradiation

Ultrasound has been reported as a tool for cell manipulation, including cell detachment [45,46,47,48]. In particular, ultrasound-based cell manipulation, using the kHz-order frequency of ultrasound in ubiquitous culture vessels, has been reported, since culture temperature is not significantly increased [49,50]. Consistent with this, cell manipulation methods, including single-cell detachment using ultrasound, have been developed for cells cultured in ubiquitous culture vessels [51,52,53]. Furthermore, ultrasound vibrations propagated with homogeneous intensity to the culture vessels provided acoustic pressure on the cells fabricating a confluent monolayer on culture surfaces, thereby resulting in the harvest of intact myoblast (C2C12) cell sheets from culture dishes and flasks [54].

Although in vivo experiments were not performed, in vitro experiments demonstrated the suitability of C2C12 cell sheets from the viewpoints of viability, protein and mRNA expression, and cell metabolism. The advantage of this method is that it does not require additional materials except ubiquitous culture vessels for cell sheet detachment. Hence, biocompatibility should be expected, and the running cost is very low. This technology can become an important method, even though further consideration is still required.

Furthermore, ultrasound was used for fabricating cell sheet-like tissue in a previous study [55]. In this research, an ultrasound standing wave was generated and cells were trapped at the nodal position of its acoustic pressure. Owing to the one-dimensional distribution of acoustic pressure, the fabricated 3D cell aggregates had the shape of a disc whose thickness was regulated by the number of seeded cells. In this experiment, the maximum thickness and diameter of the cell sheets were 2.7 mm and 8 mm, respectively.

## 3. Improving the Function of Cell Sheets

After the fabrication of cell sheets, their function should be improved for use in each application. Imitation of the physiological environment is a basic strategy employed to engineer cultured cells. Electrical and mechanical stimulation of muscle-type cells, building 3D structures, and co-culturing with other cell species can be possible strategies for imitating the physiological situation under in vitro conditions [56,57]. Figure 4 shows a summary of each method.

### 3.1. External Stimulus on Cell Sheets

After detaching cell sheets, the functions of cell sheets should be improved for each medical application by using an appropriate culture environment that mimics physiological conditions. Although the modification of media is a conventional strategy, the current methods of providing external stimuli to cells are introduced in this section. Muscle cells get excited from altered membrane potential and are exposed to contractions in our body. Hence, electric pulses and mechanical stretches have been applied to muscle-type cells in vitro [58,59]. Each of them utilizes the features of cell sheets, which are easy to handle.

#### 3.1.1. Electric Pulse

Exposing cells to an electric pulse is one of the tissue maturation methods, since muscle contracts in response to electric stimulation in vivo. By applying an electric pulse, muscle cells can contract and become exposed to mechanical stimuli produced by their own contraction. Thus, although it is unclear whether electrical stimulus itself has a huge impact on the maturation of cells or excited contraction does, many researchers have reported maturation methods using electric pulses [58,59].

Electrical stimulation has been practiced since the 20th century as a muscle-tissue maturation method, since muscle tissue that is exposed to electrical stimulation contracts under physiological conditions [60,61]. It has been reported that electrical stimulation improves the maturity of both each cell and the entire tissue in muscle tissue. Each cell shows improved functions in terms of contraction force, displacement, and speed. Furthermore, the entire tissue shows a morphological change in skeletal muscle tissue and synchronized spontaneous beating in myocardial tissue, thereby resulting in significant beating. Thus, as a result of enhancement on both the cell and tissue level, the function of the cell sheet is improved.

Electrical stimulation has also been utilized for the maturation of muscle cell sheets [62,63,64]. In cell sheets, cells not only physically but also electrically connect with each other through gap junctions. The traction force of an entire cell sheet can be measured easily, compared with that of single cells. Further, simply by overlapping two cell sheets, multiple cell sheets can be electrically connected, and their beating can be synchronized by simply culturing them together. Therefore, by taking advantage of these strong points, our group demonstrated the potential of electrical stimulus on muscle cell sheets. For skeletal muscle cell sheets, it was reported that the contraction of cell sheets coincided with electrical pulse stimulation [64]. For myocardial cell sheets, our group reported that electrical stimulation improved the function of the cell sheets. The spontaneous beating of each cell became synchronized due to the formation of an electrical connection within a cell sheet [63]. Further, the function of each cell can also be enhanced by improving the beating rate. Thus, by electric pacing, the beating rate of each cell can be regulated, and the contraction force of the entire cell sheet can be increased.

#### 3.1.2. Mechanical Stretch

Mechanical stretching, especially cyclic stretching, is known to cause the maturation of muscle cells, as reported previously [65,66]. On the other hand, the effect of single mechanical stretching was first reported on cardiac cell sheets [67]. The induced pluripotent stem-derived cardiac cell sheet was detached and transferred onto a device that stretched the cell sheets. Therefore, as a result of single stretching, the morphologies of both the entire cell sheet and each cell changed. While a cell sheet without any mechanical stress had a square shape, stretching changed the shape of cell sheets into a rectangular shape. The orientation intensity of each cell within a stretched cell sheet became significantly higher than that of the control. After stretching, the cell sheets were transplanted onto rat skeletal muscle in vivo, and even after two weeks, the cell sheets maintained their rectangular shape and orientation intensity.

In short, a novel but simple technology for myocardial tissue maturation, especially for improving the orientation intensity of myocardial fibers, was developed. Here, tissues with highly oriented myocardial fibers show stronger beating than those with the random orientation of myocardial fibers. Thus, this method should have the potential of improving the function of myocardial cell sheets. Further, although this study focused on the effect of single stretching, in other studies, the effects of cyclic stretching on muscle cells cultured in a collagen gel have been reported. Furthermore, it has already been reported that cyclic stretching on cell sheets promotes the differentiation of tendon tissues [68]. Hence, it is expected that cyclic stretching will be applied to the cell sheets, to improve their function.

### 3.2. Stack of Cell Sheets

The strategy of layering multiple cell sheets was practical to achieve 3D tissue, using cell sheet technology. The three main methods for layering cell sheets are introduced in the sections mentioned below.

#### 3.2.1. Simple Pipetting

Simple pipetting was first attempted while layering a cell sheet onto another. In this method, cell sheets are allowed to shrink owing to their intercellular tension. As a result, detached cell sheets became thicker than they were before detachment. This simple method has been used from the early days of cell sheet technology to date. For layering cell sheets, the entire cell sheet along with the media was carefully aspirated with a pipette tip and then transferred to another culture dish. To spread the cell sheets, culture media was added dropwise to the transferred cell sheet, which may be folded during transfer. To allow the cell sheet to adhere to the culture surface, the cell sheet was incubated at 37 °C, for 30 min, and became the first layer of layered cell sheets. A second layer of cell sheet can be prepared by using the same procedure as the first one, and the same procedure can be repeated to prepare new cell sheets. Note that adding pressure to the cell sheets by centrifugation promotes adhesion between the layered cell sheets, thereby resulting in an efficient 3D tissue fabrication process, as previously reported [69].

#### 3.2.2. Artificial Support Membranes

To make the procedure of cell sheet layering easier and more stable, artificial support membranes were invented. A polyvinylidene difluoride membrane, or CellShifter™, was reported to be used as a support membrane contacting the cell sheet directly [70]. The support membrane was placed on monolayered cells cultured on a temperature-responsive surface so that the cells could adhere to the membrane. Subsequently, the culture temperature was decreased while detaching the cell sheet from the culture surface. By using this method, we can recover a cell sheet with the membrane, without any shrinkage. To layer cell sheets, the cell sheet on the membrane can be placed on another culture dish with monolayered cells, and then the cell sheets are transferred from the culture dish to the membrane. Layered cell sheets can be acquired by repeating this process. Further, not only layering the cell sheet but also transplantation is a potential application of this method [71].

#### 3.2.3. Plunger-Like Manipulator

The plunger-like manipulator method was invented as the newest method for stacking cell sheets [72]. In this method, hydrogels are applied to the surface of a plunger-like manipulator, which is used as a supporting material for adhering the cell sheets. Confluent cells on the surface of a temperature-responsive culture dish can be recovered easily as a cell sheet by using the manipulator and decreasing temperature in combination, since hydrogels strongly adhere to many kinds of cells. A cell sheet can be recovered with a manipulator, without any shrinkage, using this method. Tissues with a thick 3D structure can be easily fabricated by repeating this procedure, since the tissue on the hydrogel can be easily handled and transplanted. This method should be the first candidate when considering the automation of the cell sheet stacking process. This is because the movement required in this method is quite simple compared with the other methods.

### 3.3. Co-Culture

Since most of the tissues in our body are composed of multiple cell types, co-culturing is a logical method to imitate the physiological environment in vitro. Co-culture in cell sheet technology can be classified into two concepts. One is stacking two different mono-cultured cell sheets, and the other utilizes multiple species of cells in a one-layer cell sheet.

#### 3.3.1. Stacking Two Different Mono-Cultured Cell Sheets

One of the features of the cell sheet is that it is easy to regulate the position of cells. In particular, even the vertical position of cells can be controlled easily by simply changing the order of the cell sheet layers. Taking advantage of this feature of the cell sheet technology, human umbilical vein endothelial cells (HUVECs) and normal human dermal fibroblasts (NHDFs) were co-cultured in 3D tissue fabricated with cell sheet technology [73]. In short, they prepared mono-cultured HUVEC sheets and NHDF sheets. Subsequently, triple-layered tissue, using cell sheets, was prepared, and the layers were arranged in different orders, to find the suitable order of layered cell sheets for building vascular networks in vitro. One HUVEC sheet and two NHDF sheets were layered, and the HUVEC sheet could be positioned as the top, middle, and bottom layers. Eventually, HUVECs were placed at the bottom to promote vasculogenesis. Since the location of cells can be regulated by cell sheet technology with this method, it was demonstrated that location plays an important role in regulating the function of cells and tissues.

#### 3.3.2. Co-Cultured Mono-Layered Cell Sheets

It is possible to fabricate co-cultured and mono-layered cell sheets by simply detaching co-cultured confluent cells from the culture surface. There are several methods for seeding cells, such as seeding all cells at once, varying the timing of seeding cells for each cell species, or using patterned surfaces to obtain patterned co-cultures. Several reports on the use of a co-cultured monolayer cell sheet have been reported. Since co-culture that imitates a physiological culture condition exposes cells to paracrine effects, cell sheet function can either be improved or remain similar to the tissue present in vivo. In some cases, the engraftment rate was improved by co-culturing [74]. Furthermore, co-culture is helpful for cell sheet detachment if target cell species has weak cell–cell adhesions. Co-culture with cells having a strong contraction force allows stable and efficient detachment of cell sheets [75].

It was recently reported that co-cultured cell sheets prepared from primary human hepatocytes (PHHs) and NHDFs show a unique phenomenon of the formation of ballooned hepatocytes (BHs), which are enlarged, abnormal hepatocytes [76]. This phenomenon is observed in the damaged liver, and therefore, this organ model can be used to develop drugs. Although the formation of BHs in vitro has been rarely reported, the co-cultured cell sheet showed this characteristic phenomenon. To prepare cell sheets, PHHs were seeded and cultured for 24 h on temperature-responsive surfaces, and then NHDFs were seeded on them. After 72 h of culture, co-cultured cell sheets were harvested by decreasing the temperature. In this study, the authors claim that the growth factors secreted from NHDFs and mechanical tension generated due to the 3D structure of a cell sheet are important factors for the formation of BH. Note that since the cell sheets shrink during detachment, they still have a 3D structure even if it is a monolayer cell sheet.

Patterned dual thermo-responsive surfaces were used to obtain patterned co-cultures, and a methodology to achieve patterned co-cultured cell sheets was developed [77]. It is known that the critical solution temperature at which PIPAAm can achieve a distinct transition from hydrophobic to hydrophilic could be lowered by introducing n-butyl methacrylate to PIPAAm-grafted surfaces. Thus, a surface where n-butyl methacrylate was introduced to certain areas on the PIPAAm-grafted surfaces was developed, and a dual thermo-responsive surface was achieved. On this surface, the first layer of cells was incubated at 27 °C so that the cells could adhere only to the surface grafted with n-butyl methacrylate. Subsequently, at 37 °C, the second layer of cells was seeded, and the other surface was covered with the second set of cells. After a certain duration of culture, incubation at 20 °C resulted in the detachment of a patterned co-cultured cell sheet consisting of two cell species.

## 4. Vasculogenesis in the Layered Cell Sheets

While conducting the trials of engineering cell sheets to improve their function as tissues, fabricating tissues with perfusable blood vessels is one of the most important goals. Although 3D cell-dense tissues can be easily fabricated by using cell sheet technology, the 3D cell-dense tissue that exceeds a certain thickness induces necrosis of cells inside the tissue due to ischemia [78]. The viability of the cells that compose the tissue directly affects the function of the tissue, which is, therefore, the motivation for introducing blood vessels into tissues [79]. In this section, we discuss the studies that introduced a perfusable blood vessel structure into a cell tissue fabricated by cell sheet technology, to maintain cell viability. In these studies, animal models, animal tissues, and artificial scaffolds were utilized to build perfusable vessel structures. These three methods are categorized into in vivo, ex vivo, and in vitro methods. These strategies have their own characteristics, as described in Figure 5.

### 4.1. In Vivo Method

Perfusable blood vessels introduced in a layered cell sheet to overcome ischemia occurring in cell-dense tissue were first reported in a study that used an animal model as a living in vivo bioreactor [80]. Layered myocardial cell sheets detached from thermo-responsive culture surfaces were transplanted onto rat dorsal subcutaneous tissues, and it was observed that only three layers could be grafted owing to ischemic injury of the transplants. On the other hand, when a cell sheet construct having less than three layers was transplanted, a well-organized vascular network was created in vivo after a few days. Based on this observation, the authors suggested that repeated transplantation of layered cell sheets after a few days can be useful for achieving organized vascular networks and overcoming the limitations of layered grafts. Furthermore, since both vascularization and electrical connections between overlaid triple-layer grafts were observed, spontaneous pulsation between the two transplanted grafts at one-, two-, or three-day intervals after the transplantations was observed. One month after the second graft implant procedure, two grafts spontaneously pulsated and their pulsations were completely synchronized when the grafts were transplanted at one- and two-day intervals. Further, a one-day interval allowed two grafts to be perfectly attached to each other due to the absence of in-grown connective tissue, which was a unique event that occurred in vivo. Thus, one- and two-day intervals were used in the polysurgery of cell sheet grafts. Spontaneous and synchronized graft contraction/relaxation was clearly visible from outside the rat after repeating the surgery 10 times. Further, the thick tissue developed by using the polysurgery method had well-organized micro-vessels, and the thickness of the tissue was almost 1 mm. This study used live animal models as an in vivo bioreactor for the first time to overcome the limitation of the thickness of cell-dense tissue due to ischemia, which showed that the development of vascular networks could be a good strategy to fabricate thick cell tissues.

### 4.2. Ex Vivo Method

As a next step, a method of in vitro vasculogenesis in cell-dense tissue has been investigated. During many trials, co-culture with endothelial cells (ECs) was identified as an important factor for inducing vasculogenesis in vitro. However, only EC networks with partial tubular structures have been produced. On the other hand, perfusable EC networks were identified in layered myocardial cell sheets during in vitro culture with a bioreactor, using rat femoral tissue as a vascular bed [81]. In short, this method should be referred to as the ex vivo method. Triple-layered myocardial/EC cell sheets were cultured on a resected section of rat femoral tissue with a connectable artery and vein, which was then perfused with culture medium in vitro. To achieve an adequate culture environment to promote vasculogenesis, fibroblast growth factor 2 was added to the perfused culture medium. When culturing triple-layered EC/myocardial cell co-cultured sheets on the vascular bed for three days, ECs in the cell sheet migrated into the vascular bed, and it was found that ECs contributed to the fabrication of perfusable vascular networks.

To scale-up the layered co-cultured cell sheet, the triple-layer sheet was repeatedly layered on the vascular bed and perfused with the culture media. A second triple-layer tissue was placed on the first tissue after three-day culture and then perfused for another three days. Subsequently, after repeating the procedure multiple times, thicker and more viable cell tissues were acquired than those obtained with a single-step procedure, where six cell sheets were layered at once with/without the vascular bed. Although the three-day culture takes longer than the method introduced in the previous section, there was no chance for connective tissue invasion to occur in this environment. Further, to demonstrate the potential of the present method in transplantation, the six-layer tissue developed by using the proposed multiple procedure with/without vessel anastomoses was transplanted. In grafts with vessel anastomoses, the vascular structures in the grafts were maintained, and the grafts beat for two weeks after the procedure. This indicates the potential of a method that uses animal tissues as a vascular bed in an ex vivo bioreactor for practical treatment.

### 4.3. In Vitro Method

Although the idea we would like to introduce in this section might have been similar to the method described in the previous section, this method used an artificial vascular bed made of collagen gel [82]. This study demonstrated the effectiveness of a vascular bed made with an artificial collagen gel to induce vascularization, and this concept can be utilized in other systems when performing similar experiments [83,84].

It has been reported that collagen-based vascular beds have micro-channels that imitate vascular networks [82]. Triple-layered myocardial/EC cell sheets were cultured on an artificial vascular bed set in and connected to a bioreactor that perfused culture media containing vascular endothelial growth factor (VEGF) and basic fibroblast growth factor (bFGF) into the micro-channels. ECs in layered sheets migrate into the vascular bed and connect with the micro-channels. The culture medium was perfused into the layered sheets from the micro-channel of the vascular bed, and new capillaries were developed in the layered sheets.

Thus, to scale-up layered cell sheets, they were repeatedly layered on the vascular bed and perfusion was continued. A triple-layer tissue was cultured for five days, with continuous perfusion, and then a second triple-layer tissue was placed on the first tissue, and they were continuously cultured for another five days, in the bioreactor, with perfusion. This procedure generated thicker and more viable cell tissues than those obtained with a single-step procedure layering six cell sheets at once on the vascular bed. This layering and five-day culture process could be repeated to make 12-layer cell sheets having thicknesses greater than 110 µm. Although the 12-layer cell sheets could cause cell death focally, each layer adhered physically and electrically to each other. Furthermore, since no animal cells or tissue were used for perfusion in this method, the fabricated tissue may be suitable for clinical study. The key results provided critical insights into the strategies employed for developing clinically important tissue surrogates with in vitro bioreactor designs.

## 5. Conclusions

In this review, we summarized the reported techniques employed for developing cell sheet technology. Since cell sheet technology has demonstrated its potential in regenerative therapy, many researchers have developed methods for fabricating cell sheets based on their own technologies and ideas. Based on this review, users of cell sheets can employ a suitable method considering its advantages. The technologies used to improve the functions of cell sheets were also reviewed. The fact that a variety of methods for harvesting cell sheet have been developed shows the usefulness and advantage of cell sheets in tissue engineering research field. Although there are many methods, one basic idea for improving the function of cell sheets in vitro is to imitate the physiological environment. In this step, each method was attempted individually. Combining multiple methods can be useful for enhancing the function of the cell sheet; however, it must depend on the cell species employed for the fabrication of the cell sheets. Finally, we reviewed our strategies for vasculogenesis to overcome one of the biggest challenges in cell sheet technology that causes cell death owing to ischemia. As a prospective applications of cell sheet technology, many options, such as tissue engineering, regenerative medicine, and bio-actuator, were suggested by many researchers [8,85,86]. In any case, fundamental technology for cell-sheet-based tissue engineering should encourage every challenge. Thus, in this paper, we reviewed the methodologies used for developing cell sheet technology, and we hope that collaborations will be proposed by the readers of this review.

## Figures and Tables

**Figure 1 ijms-22-00425-f001:**
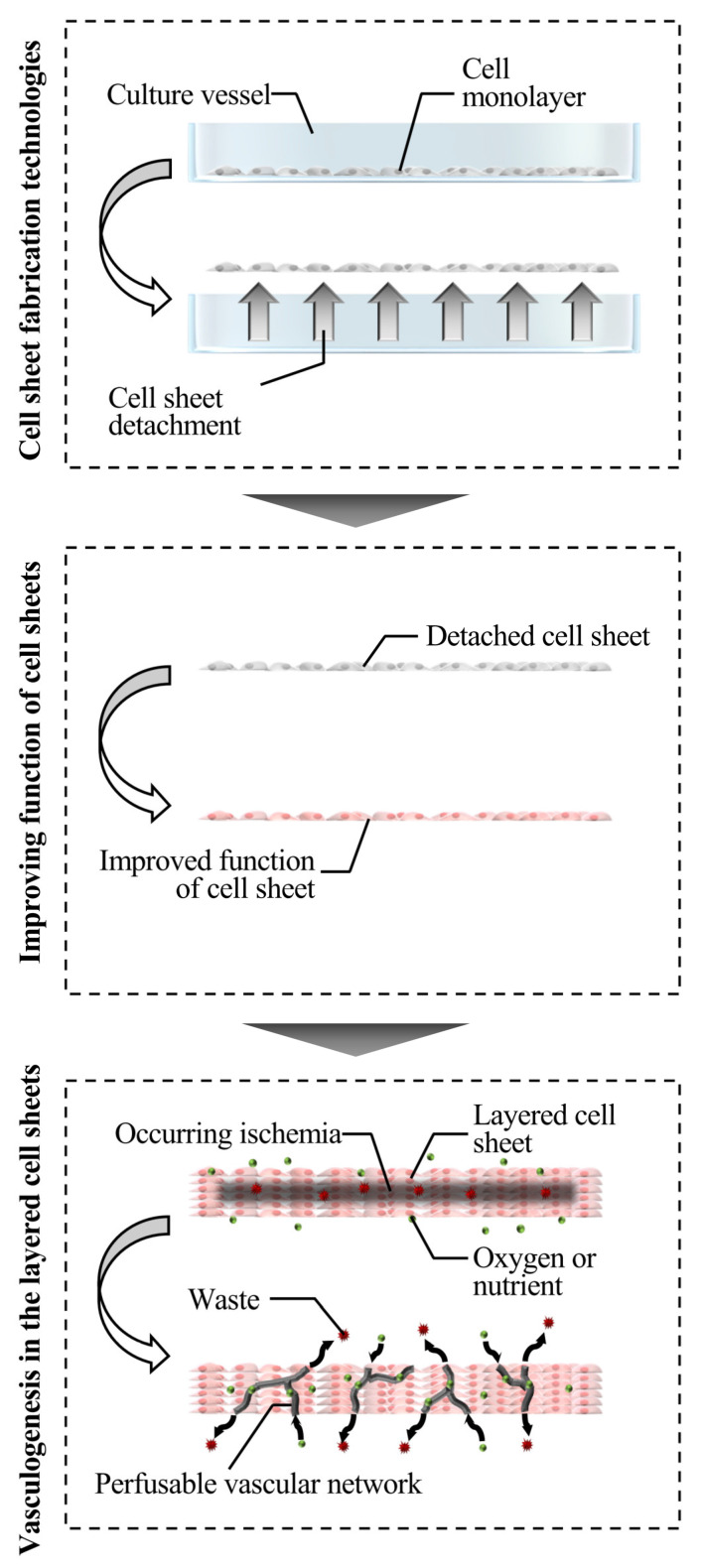
Schematic diagram that categorizes various technologies employed for developing cell sheet technology discussed in this review.

**Figure 2 ijms-22-00425-f002:**
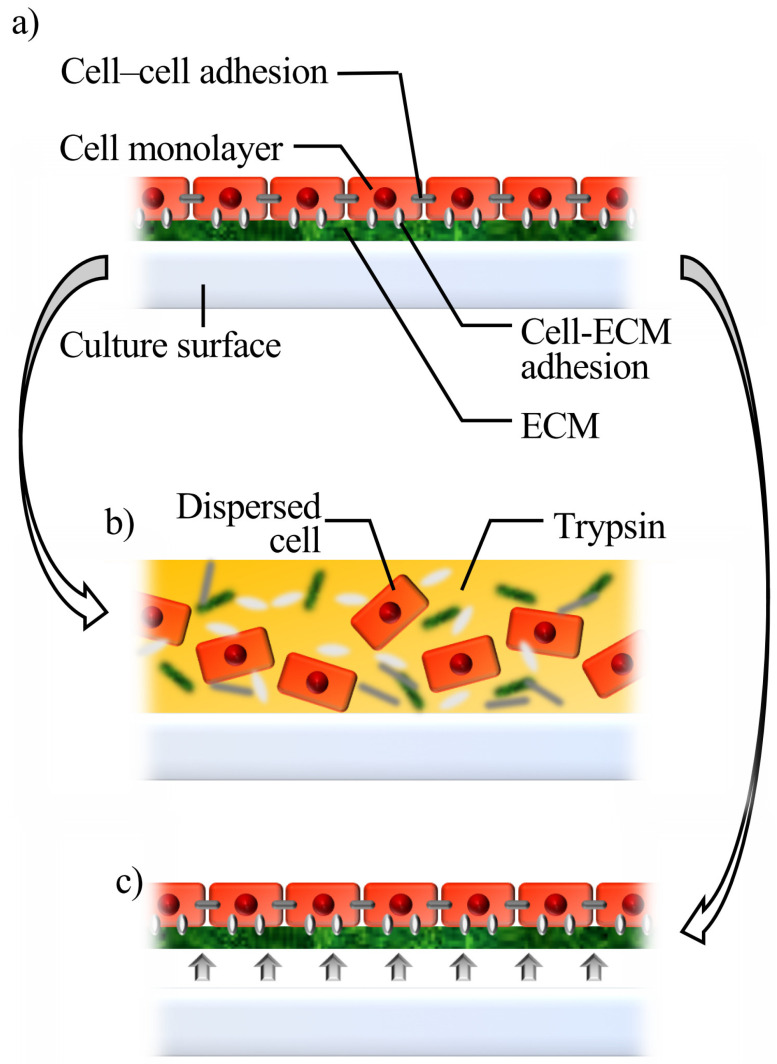
Schematic diagram of cell sheet fabrication. (**a**) Confluent cells on a culture surface. (**b**) Trypsin digests extracellular matrix (ECM) and other surface proteins. Thus, cells become dispersed and lose proteins that are essential for maintaining cell functions. (**c**) When adhesions between the cell-culture surface are selectively lost, confluent cells can be detached in the shape of a sheet.

**Figure 3 ijms-22-00425-f003:**
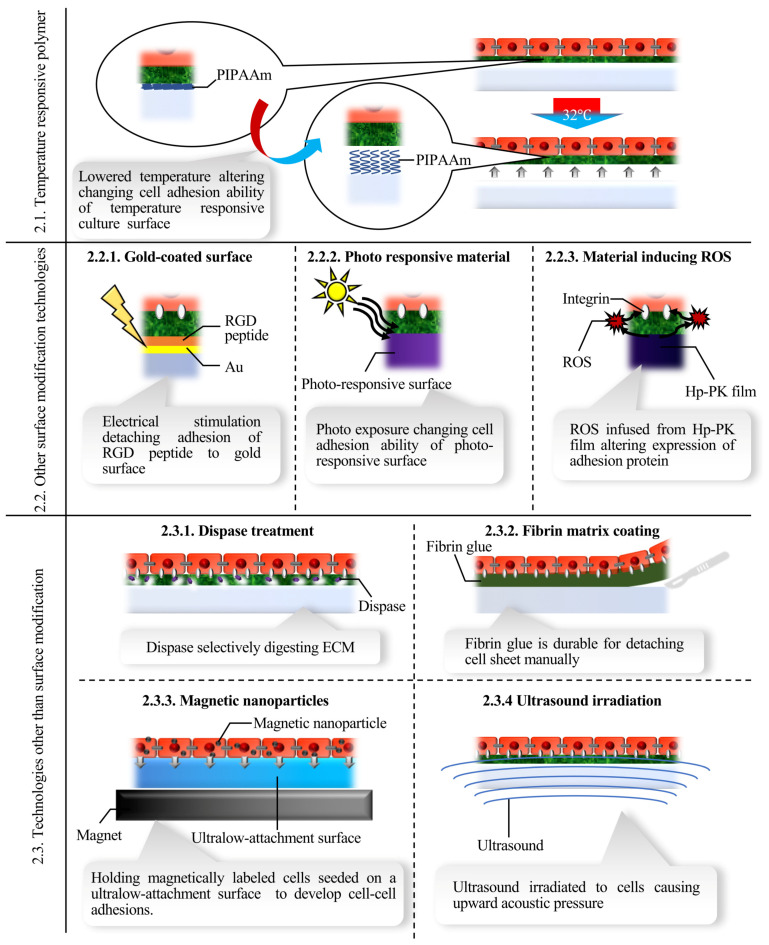
Summary of each cell sheet fabrication method introduced in the present review. PIPAAm, poly (N-isopropylacrylamide); RGD, Arg–Gly–Asp; Au, gold; ROS; reactive oxygen species; Hp-PK, hematoporphyrin-incorporated polyketone; ECM, extracellular matrix.

**Figure 4 ijms-22-00425-f004:**
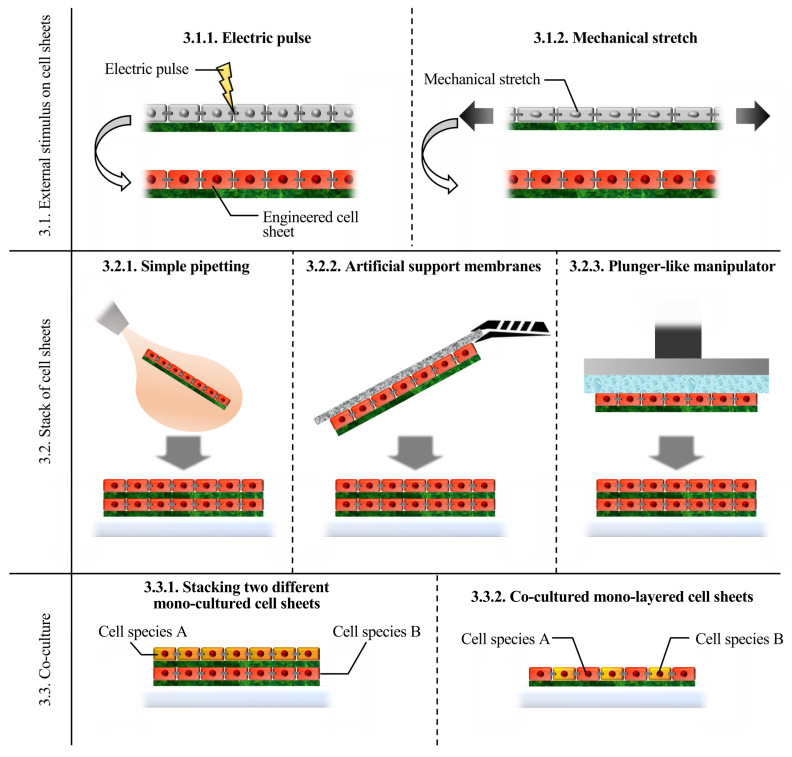
Summary of each cell sheet engineering method introduced in the present review.

**Figure 5 ijms-22-00425-f005:**
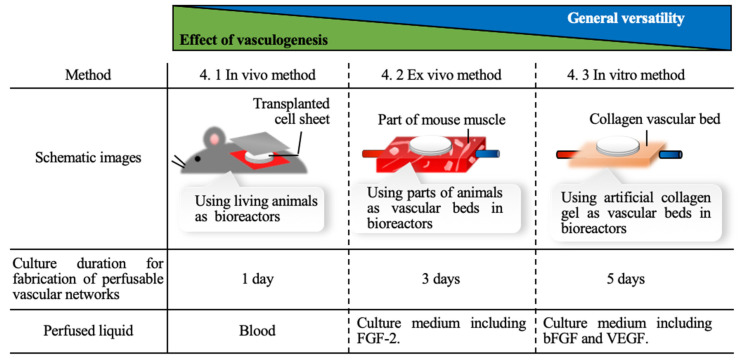
Comparison of each strategy applied for building perfusable vascular network that is described in this review. FGF-2, fibroblast growth factor 2; bFGF, basic fibroblast growth factor; VEGF, vascular endothelial growth factor.

**Table 1 ijms-22-00425-t001:** The features of each method.

		Feature	Special Equipment	Triger of Detachment	Possible Residual Substances
2.1. Temperature responsive polymer		Commercially available and clinically reliable.	Low-temperature incubator	Temperature reduction	-
2.2. Other surface modification technologies	2.2.1. Gold-coated surface	Implementation of 3D-shaped cell sheets.	Potentiostat	Electric stimulus	RGD peptide
	2.2.2. Photo responsive material	Good temporal and spatial controllability.	Illumination device	Photo exposure	-
	2.2.3. Material inducing ROS	The ease of transplantation owing to flexible substrate.	Illumination device	Photo exposure	ROS
2.3. Technologies other than surface modification	2.3.1. Dispase treatment	Simple method to just add an enzyme.	-	Adding enzyme	Dispase
	2.3.2. Fibrin matrix coating	Long-term durability for handling and good adhesion properties	-	Manual handling	Fibrin
	2.3.3. Magnetic nanoparticles	Potential in regenerative therapy was confirmed by in vivo experiments	Magnet	Remove of magnetic field	Magnetic nanoparticles
	2.3.4. Ultrasound irradiation	High biocompatibility and low running cost	Ultrasound transducer	Ultrasound irradiation	-

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
