# Peer review of "Fundamental Technologies and Recent Advances of Cell-Sheet-Based Tissue Engineering"

_ijms, 2021, doi:10.3390/ijms22010425_

Round 1

Reviewer 1 Report

Comments to Authors: The topic is interesting but the authors need to improve the work in order to justify the novelty.

Specific Comments

  1. Figure 1 is blank. Please verify
  2. Where is Figure 2?
  3. The need for cell-sheet technology must be discussed & included in the introduction part
  4. In section 2.1 Temperature-responsive culture surface, not just PIPAAm, but several others polymers or combinations such as PIPAAm(PEG)-PM, P(IPAAm-co-CIPAAm) etc have also been developed. Hence these must be included.
  5. This section can also include a table on the various materials/polymers used in the Temperature-responsive method also indicating the LCST or cell sheet detachment temperature and application.
  6. The title “Recent advances of cell sheet technology in regenerative therapy and tissue model fabrication” is not at all justified. Although different fabrication approaches have been discussed, their application aspects & tissues-of-interest have not been included. Or else the title has to be modified.
  7. The authors can include a table indicating the advantages & limitations of each fabrication method.
  8. The challenges in applying cell sheet technology are essential and a section on the same must be included.
  9. The prospective applications of cell sheet technology can also be suggested.
  10. Similar review articles have already been published [Lu, Y., Zhang, W., Wang, J. et al. Recent advances in cell sheet technology for bone and cartilage regeneration: from preparation to application. Int J Oral Sci 11, 17 (2019); Cell sheet technology: a promising strategy in regenerative medicine, Cytotherapy Volume 21, Issue 1, January 2019, Pages 3-16]. So what’s the novelty of the current manuscript?

Reviewer 2 Report

The review article of “Recent advances of cell sheet technology in regenerative therapy and tissue model fabrication” presented by Chikahiro Imashiro and Tatsuya Shimizu elaborately reviewed the cell sheet technology in a scientific manner.  The manuscript was scientifically and comprehensively describing all the background information of cell sheet technology in regenerative therapy. I recommend the manuscript for minor revision and to be published in your esteem journal.

Comments

  1. Overall, the paper is easily read, but there are still some minor problems in language and spelling. A throughout checking is recommended.
  2. It should be advised to check the all the abbreviation cited in the text.
  3. The abstract needs revision, it should be highlighting the prospects of the study.
  4. The paragraph of ‘Cell sheet fabrication technologies’ (Page No: 2; Line No: 10-19) was hard to understand. it will be modified.
  5. Check the Figure 1, the figure scheme was quite confusing, I guess the figure content were missing.
  6. Page No: 8; Line No: 11: These sentences are meaningless “we introduced several approaches to improve the function of cell sheets for their application.
  7. If so possible ‘Improving the function of cell sheets’ might be improved with pictorial representation.

Round 2

Reviewer 1 Report

The manuscript looks much better and improved.

Author Response

Thank you for your huge contribution to our manuscript.